# Ethnobotanical survey of medicinal plants used by indigenous knowledge holders to manage healthcare needs in children

**Peter Tshepiso Ndhlovu**[1,2], **John Awungnjia Asong**[3], **Abiodun Olusola Omotayo**[4], **Wilfred Otang-Mbeng**[2], **Adeyemi Oladapo Aremu**[1,4,5] *

**1** Indigenous Knowledge Systems (IKS) Centre, Faculty of Natural and Agricultural Sciences, North-West University, Mmabatho, South Africa, **2** School of Biology and Environmental Sciences, Faculty of Agriculture and Natural Sciences, University of Mpumalanga, Mbombela, South Africa, **3** Unit for Environmental Sciences and Management, Faculty of Natural and Agricultural Sciences, North-West University, Potchefstroom, South Africa, **4** Food Security and Safety Area Research Group, Faculty of Natural and Agricultural Sciences, North-West University, Mmabatho, South Africa, **5** School of Life Sciences, College of Agriculture, Engineering and Science, University of KwaZulu-Natal (Westville Campus), Durban, KwaZulu-Natal, South Africa

* Oladapo.aremu@nwu.ac.za

**Data Availability Statement:** All relevant data are within the paper.

**Funding:** PTN is received funding from the National Research Foundation (NRF Thuthuka grant UID:

## Abstract

Childhood diseases remain an increasing health problem in many developing countries and also associated with an enormous financial burden. In South Africa, many people still rely on traditional medicine for their primary healthcare. However, inadequate documentation of medicinal plants used to manage childhood diseases remain a prominent gap. Hence, the current study explored the importance of medicinal plants to treat and manage childhood diseases in the North West Province, South Africa. An ethnobotanical survey was conducted with 101 participants using semi-structured interviews (face-to-face). Ethnobotanical indices such as Frequency of citation (FC), Use-value (UV) and Informed Consensus Factor (ICF) were used for data analysis. A total of 61 plants from 34 families were recorded as medicine used for managing seven (7) categories of diseases resulting from 29 sub-categories. Skin-related and gastro-intestinal diseases were the most prevalent childhood health conditions encountered by the study participants. Based on their FC values that ranged from approximately 0.9–75%, the most popular medicinal plants used by the participants were *Aptosinum elongatum* (75.2%), *Commelina diffusa* (45.5%), *Euphorbia prostrata* (31.6%) and *Bulbine frutescens* (31.7%). In terms of the UV, *A. elongatum* (0.75), *C. diffusa* (0.45), *E. prostrata* (0.31), *H. hemerocallidea* (0.19) and *E. elephantina* (0.19) were the dominant plants used for treating and managing childhood diseases. Based on ICF, skin-related diseases dominated with the highest ICF value of 0.99. This category had 381 use-reports, comprising 34 plants (55.7% of total plants) used for childhood-related diseases. Particularly, *B. frutescens* and *E. elephantina* were the most-cited plants for the aforementioned category. Leaves (23%) and roots (23%) were the most frequently used plant parts. Decoctions and maceration were the main preparation methods, and the plant remedies were mainly administered orally (60%) and topically (39%). The current study revealed the continuous dependence on the plant for primary health care relating to childhood diseases

138298) Pretoria, South Africa. We appreciate additional financial support from the Higher Degree Committee (HDC) of the Faculty of Natural and Agricultural Sciences, North-West University and the University of Mpumalanga, South Africa. The funders had no role in study design, data collection and analysis, decision to publish, or preparation of the manuscript.

**Competing interests:** The authors have declared that no competing interests exist.

in the study area. We generated a valuable inventory of medicinal plants and associated indigenous knowledge for child healthcare needs. However, investigating the biological efficacies, phytochemical profiles and the safety of these identified plants in relevant test systems remain essential in future research.

## 1. Introduction

As common with many middle-income countries, South Africa is still facing a high child/infant mortality rate [1]. Even though the current new-born mortality rates are within the United Nations Sustainable Development Goal (SDG-3) target of 12 deaths per 1000 live births, the absolute number of deaths is unacceptably high for South Africa [2]. The current infant mortality rate for South Africa in the first quarter of 2022 was 24.3 deaths per 1000 live births, which is a 2.93% decline from 2021 [3]. The overwhelming tide of emerging epidemics and pandemics has increased the strain on the healthcare system. It has also affected resources often allocated for the prevention of diseases and the active promotion of children health, especially in rural areas [4–6]. Even though South Africa has several databases that collect information on neonatal deaths and prevalent diseases, most of these focus on deaths occurring within healthcare facilities [2]. Outside such facilities, the number of deaths is largely unknown, child mortality, compared to the 12 per 1000 reported from the District Health Information System data [7].

Globally, the existence of traditional medicine depends fundamentally on the rich diversity of plants and the related knowledge of their use as herbal therapy [8]. These medicinal plants remain indispensable for maintaining health and well-being among several ethnic groups [9–11]. The on-going dire economic situations in most rural areas and limited access to conventional medicine have increased the popularity of traditional medicine [12]. In addition, the use of medicinal plants for childhood diseases in rural areas have been receiving an increased attention among researchers [13–15].

South Africa is endowed with a rich wealth of flora and is often acclaimed as a biodiversity hotspot [16]. An estimated 30000 plants are used for traditional medicine for the management of diverse health conditions [13, 17, 18]. In South Africa, the significance of medicinal plants as remedies against many diseases among local communities are well-recognised [19–22]. However, the need to develop a comprehensive database for medicinal plants used for the management and treatment of childhood diseases cannot be overemphasized. In addition, the increasing rate of urbanization and habitat destruction that has a detrimental effect on plant resources especially medicinal plants, remain a global challenge [23]. A recent review by Ndhlovu et al. [13] revealed the dearth of scientific investigation on medicinal plants used to treat and manage childhood disease in North West Province of South Africa. Thus, the current study explored the indigenous knowledge and medicinal plants used to manage and treat childhood diseases among local communities in North West Province, South Africa.

## 2. Materials and methods

### 2.1. Description of the study area

The study was conducted in 17 selected communities in the Ngaka Modiri Molema and Bojanala districts in the North West Province, South Africa (Fig 1 and Table 1). We prepared the map of the study area using the free software QGIS version 3.22.14 Bilowieza (available at: www.qgis.org).

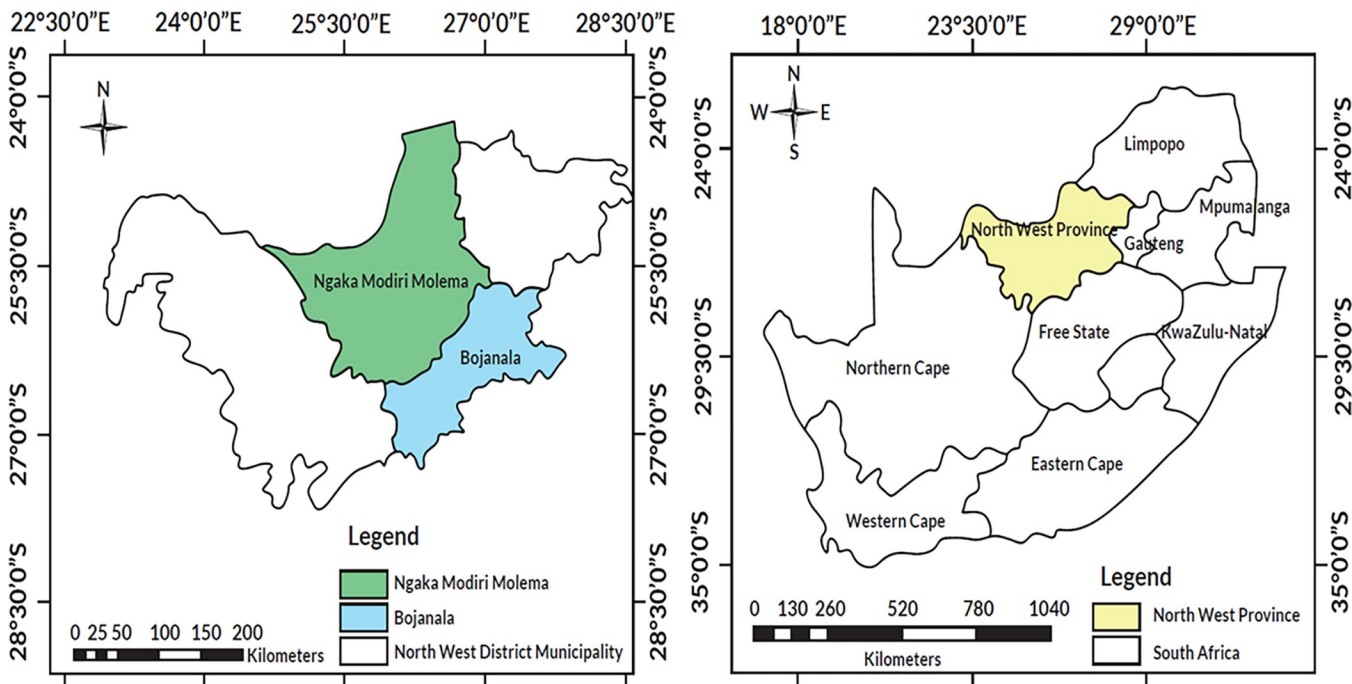

**Fig 1. Selected district municipalities (Ngaka Modiri Molema and Bojanala) in the North West Province, South Africa.** The map was prepared using the free software QGIS version 3.22.14 Bilowieza (available at: www.qgis.org). This figure is licensed under Creative Commons Attribution License (CCAL), CC BY 4.0.

The selected district lies between latitude 22° South and longitude 28° East of the North West Province, covering 116 320 km$^2$ which is about 9.5% of the total surface area of South Africa. North West Province shares boundaries with four other provinces, namely Northern Cape, Free State, Gauteng and Limpopo provinces [24, 25]. The average annual rainfall is about 360 mm, mostly experienced in the summer months between October and April, while

**Table 1. Selected study sites (villages and towns) in Ngaka Modiri Molema and Bojanala districts of North West Province, South Africa.**

| Ngaka Modiri Molema district | Co-ordinates | Bojanala district | Co-ordinates |
|---|---|---|---|
| 1. Mahikeng | 25° 51' 11.3796" S<br>25° 38' 24.6516" E | 1. Brits | 25°39'1.18"S<br>27°48'8.35"E |
| 2. Disaneng | 25°48'33.29"S<br>25°15'36.18"E | 2. Kgabalatsane | 25°32'9.8"S<br>27°57'25.62"E |
| 3. Tsetse | 25°43'53"S<br>25°40'12.82"E | 3. Rockville | 25°37'4.73"S<br>27°56'32.49"E |
| 4. Lotlhakane | 26°0'41.62"S<br>25°37'17.01"E | 4. Iterelelng | 25°35'10.44"S<br>28°6'7.06"E |
| 5. 'Bethel | -26.2913° E<br>26° 17' 29" S | 5. Hebron | 25°35'10.44"S<br>28°6'7.06"E |
| 6. Ramatlabama | 25°38'14.56"S<br>25°32'50.58"E | 6. Rabokhala | 25°29'25.09"S<br>27°50'29.27"E |
| 7. Seweding | 25°52'14.83"S<br>25°35'41.11"E | 7. Kilpgat | 25°31'2.14"S<br>28°1'26.91"E |
| 8. Makhunung | 25°38'55.39"S<br>25°34'31.38"E | 8. Madidi | 25°26'46.1"S<br>27°56'43.45"E |
| 9. Rramosadi | 25°51'26.63"S<br>25°36'32.16"E | | |

summer temperatures range from 17 to 31˚C and winter temperature ranges from 3 to 21˚C [26]. Ngaka Modiri Molema and Bojanala Platinum districts were selected due to their high biodiversity and economic activities. Furthermore, the population consist of 94% Black Africans, and Setswana is the most spoken language.

North West Province is one of nine provinces in South Africa and is an important contributor to the South African economy mainly through agricultural and mining activities [27, 28]. According to Stats SA [28], the public health system in North West Province has been in a state of crisis for many years. Particularly, the persistent and widespread medicine stock-outs and shortages remain common in the province. The limited supply of medicine and other pharmaceuticals has resulted in patients visiting the facilities for health care. Furthermore, most of health facilities were built on racial lines during the apartheid era. As a result, there are still challenges of inequitable distribution of health facilities and human capital resources, with a skewed spread of medical officers, professional nurses and allied health professionals across the districts [25].

## 2.2 Data collection

An ethnobotanical survey was conducted between autumn and summer from April to August 2021. In addition, from April 2021-April 2022, follow-up trips were conducted with the key participants. This was done to keep updating participants and to explore whether they could add new data to enrich the study. Prior to the collection of ethnobotanical data, an overview of the focus and the significance of the study was shared with the participants before obtaining their consent to participate. After the engagement, their consent to participate was requested for the study. A total of 101 participants were interviewed in two districts of the North West Province. The data was collected through semi-structured interviews (face-to-face) using Setswana, a widely spoken local language in the study area. These semi-structured interviews were designed to record information about the prevalence of diseases, plants used to manage and treat childhood diseases, plant parts used, and methods of preparation and administration. As a visual aid to assist with identifying diseases among the participants, photographs of known childhood diseases were extracted and compiled from reliable sources [29, 30]. This assisted the participants with the identification of childhood diseases during the survey.

## 2.3 Sampling technique

The study involved 101 participants such as traditional health practitioners (specifically those with expertise in managing and treating diseases among children) and herbal vendors in the selected study areas. Participants over 18 years old were selected because the research relies only on adults and knowledge experts. Purposive expert sampling was an advantageous technique because of the ability to generate greater knowledge depth in the field of interest [31, 32]. In addition, participants were selected based on their ability to speak and understand Setswana or English. According to Stats SA [28], both languages are the two main languages on the study sites. The participants of both genders registered with the North West Dingaka Association (regulatory body in the province), whose names also appear in the provincial Traditional Health Practitioners (THPs) database, were targeted. The participants who met the eligibility criteria of the various categories of THPs as defined in the Traditional Health Practitioners Act 22 of 2007 included Traditional Doctors, Diviners (Sangomas), Traditional Surgeons, Traditional Birth Attendants (TBAs) and Herbalists [33].

The study excluded individuals under 18 years as they are considered minors in South Africa and cannot give consent on their own [34]. Due to limited resources, participants from Dr Kenneth Kaunda and Dr Ruth Segomotsi Mopati of the North West Province were

excluded. Furthermore, the participants who were not registered with the North West Dingaka Association were excluded. Creswell and Creswell [35] indicated that the exclusion criteria include factors or characteristics that make the recruited population ineligible for a study. Exclusion requirements are a series of predefined meanings used to classify participants who will not be included or will have to withdraw from the research study after they have been included [36].

Upon the agreement by a participants, a digital voice recorder was used to capture the interviews and create an audio pool of information. In addition, photographs of each of the plant species were captured. For botanical identification, voucher specimens for all the plants were collected during the field study and deposited in the herbarium of the South African National Biodiversity Institute (SANBI), Pretoria, South Africa. Furthermore, photos were taken for identification as a conservation mechanism to prevent or reduce the risk of extinction for plants considered as threatened or facing extinction. Botanical names of the collected plant species were identified by an expert SANBI using a detailed regional dichotomous key [37].

## 2.4 Data analysis

Descriptive statistics (mean, frequency and percentage) were used to identify and describe the socio-demographic parameters of the participants [38]. Data were computed and entered into an Excel office sheet, 2016, then exported into Statistical Package for Social Sciences (SPSS version 27). Based on the previous ethnobotanical indices [39, 40], three quantitative parameters: frequency of citation (FC), use-value (UV), and informant consensus factor (ICF), was used to analyse the data.

**2.4.1 Frequency of citation (FC).** Based on the study of Trotter and Logan [41], the frequency of citation (FC) of the plant species was calculated as follows:

$$FC = \frac{Np}{N} \text{ x } 100 \tag{1}$$

Where Np = number of times a particular species was mentioned; N = total number of times that all species were mentioned x 100

**2.4.2 Use-value (UV).** The use-value of a plant species is a measure of the relative importance of how the species is known locally. It ranks the plants according to the number of uses mentioned for a particular species and the number of participants who mentioned the use of the species [39, 40, 42]. This was calculated using the formula below

$$UV = \Sigma \frac{Ui}{N} \tag{2}$$

where: UV = use-value of a species; U = number of citations per species; n = number of informants. When the UV is high (closer to 1), it means the species is highly used as medicine by the participants, and when it is very low (closer to 0), it means the plant has a few medicinal uses reported by the participants.

**2.4.3 Informed consensus factor (ICF).** This value was calculated for the different categories of diseases to ascertain the degree of homogeneity of the knowledge among the participants in the study area with respect to the treatment of childhood diseases and the use of plants per disease category [43]. The ICF was calculated using the formula below:

$$ICF = \frac{Nur - Nt}{Nur - 1} \tag{3}$$

Where $N_{ur}$ = number of use citations in each category; $n_t$ = number of plants used. All citations were placed into one of the seven categories: Pain and inflammation-related diseases, gastro-intestinal related diseases, oral-related diseases, skin-related diseases, respiratory-related diseases, urinary-related diseases and social/nutritional related diseases.

## 2.5 Ethical approval

Prior to conducting the study, ethical clearance (NWU-00485-20-A1) was obtained from the North-West University Health Research Ethics Committee (NWU-HREC), while the permit to collect plant species was obtained from the North West Department of Economic Development, Environment, Conservation and Tourism (ID NW 27370/10/2020).

The local authorities, including traditional leaders (Barolong bo Rra Tshidi), granted us the permission to access the study sites, while the participants signed the informed consent form. This study focused on the knowledge holders who were over the age of 18 and were able to give consent, as the South African laws recognized a person over the age of 18 as an adult. The informed consent form described the terms and conditions that both the researcher and participants needed to follow. For instance, the names of the participants remaining private or anonymous throughout the study, the right to withdraw whenever they feel uncomfortable proceeding with the interview. The ethnobotanical survey were conducted in accordance with the Declaration of Helsinki.

In compliance with the Nagoya Protocol on Access to Genetic Resources and the Fair and Equitable Sharing of Benefits Arising from their Utilization to the Convention on Biological Diversity, the study participants from the 17 selected communities in the Ngaka Modiri Molema and Bojanala districts retain the authorship of traditional knowledge documented in this publication. Therefore, any use of the documented information, other than for scientific publications, requires prior consent of the traditional knowledge holders and their agreement on access to benefits resulting from any commercial use.

## 3. Results and discussion

Traditional healing using medicinal plants as the core ingredient is well-enriched among many African communities, including the Batswana people [44, 45]. In South Africa, remote areas previously disadvantaged during the former political systems have limited access to modern medicine [28]. Medications, especially for children health problems, are sparingly available to worsen the situation. The dependance and reliability on plants in traditional medicine are unquestionable and taken into cognizance in the current study.

### 3.1 Socio-economic characteristics

Table 2 summaries the demographic characteristics of the participants, including THPs and herbal vendors. Gender distribution among the participants was 78% female and 21% male. This signifies the importance of females as active custodians of indigenous knowledge related to childhood health needs. This is also an indication that females play an essential role in managing households in the North West Province. In a similar manner, finding by Omotayo et al. [46] reported that the dominance (56%) of female-headed households in the North West Province. In terms of marital status, the majority (53.5%) were single, while 2% of the participants were widowed.

In the current study, the formal education status was high among the knowledgeable participants about herbal-based remedies used to treat and manage childhood diseases. For example, 63% of the participants completed a secondary level of education, 21.8% did not receive formal education and 5% attended primary school. Education plays a crucial role in the socio-

**Table 2. Demography of the participants with knowledge on child-related diseases in Ngaka Modiri Molema and Bojanala districts of North West Province, South Africa (n = 101).**

| Parameter | Frequency | Percentage (%) |
|---|---|---|
| **Age (years)** | | |
| 17–37 | 21 | 20.8 |
| 38–58 | 53 | 52.5 |
| 59–79 | 27 | 26.7 |
| **Gender** | | |
| Male | 22 | 21.8 |
| Female | 79 | 78.2 |
| **Marital status** | | |
| Married | 35 | 34.7 |
| Single | 54 | 53.5 |
| Divorced | 2 | 1.9 |
| Widowed | 8 | 7.9 |
| Other(s) | 2 | 1.9 |
| **Educational status** | | |
| Informal education | 22 | 21.8 |
| Primary education | 5 | 5.0 |
| Standard education | 64 | 63.3 |
| Post-graduate level | 10 | 9.9 |
| **Household size** | | |
| Less than or equal to four | 57 | 56.4 |
| More than four | 44 | 43.6 |
| **Type of location** | | |
| Villages | 80 | 79.2 |
| Township | 16 | 15.8 |
| Urban area | 5 | 4.9 |
| **Type of practice** | | |
| Diviner | 13 | 12.8 |
| Herbalist | 24 | 23.9 |
| Herbalist and diviner | 38 | 37.6 |
| Herbal vendor | 20 | 19.8 |
| Traditional birth attendant | 6 | 5.9 |

economic status of participants and their consciousness of environmental and health-related issues [47–49].

In addition, the household size of the participants revealed that the majority (56%) consisted of less than 5 individuals (Table 2). These findings are consistent with Stats SA [28] and Nkonki-Mandleni et al. [50], which reported an average household size of three people in 2011 and 2016, with a decrease in household size due to a lack of employment and urban migration [27]. Although 79% of the participants resided in villages, 15.8% were based in urban areas. A greater proportion of the participants with knowledge on childhood diseases were THPs such as diviners and herbalists.

## 3.2 Overview of the generated plant inventory used for treating childhood diseases

The use of plants continue to play a vital role in treating childhood diseases in South Africa, especially in rural areas [13]. In this study, 61 plant species belonging to 34 families were

recorded as herbal remedies to treat and manage 29 childhood diseases (Table 3; Fig 2). The number of plants used for childhood diseases was relatively higher compared to 44 plants documented in Nigeria [14] and 50 plants in Uganda [51].

Asteraceae (7) and Fabaceae (7) were the most dominant plant families, followed by Solanaceae (4) and Asparagaceae (4), while each of the remaining 30 families were represented by 1–3 plants per family (Fig 2). The dominant botanical families especially the Fabaceae, are known to have the highest number of species and also have the highest number of medicinal plant families in the North West Province [52]. The dominance of Fabaceae as one of the most preferred plant families has been similarly recorded in many ethnobotanical surveys in South Africa [53–55]. In Southern Africa, the importance of Asteraceae is well-documented [56]. Members of Asteraceae are widely recognized as weeds in anthropogenic contexts and are among the first species to sprout in the field after the soil has been prepared for planting [57–60]. Furthermore, Asteraceae is the most varied and cosmopolitan family of flowering plants, and they are good sources of inulin, a natural polysaccharide with prebiotic solid properties and potent antioxidant, anti-inflammatory, and antimicrobial activity [59, 61]. This could explain the high number of citations for plants in this family in many ethnobotanical studies in rural communities, where these plants from Asteraceae are frequently used in traditional medicine. Several studies have suggested that using plant species from the aforementioned families may be related to the effectiveness of bioactive ingredients against diseases and active biological compounds [42, 60].

## 3.3 Frequency of citations and use-value pattern for the documented plants

The FC values ranged from approximately 0.9–75%, and the top 10 most frequently cited medicinal plants used by the participants were *Aptosinum elongatum*, *Euphorbia prostrata*, *Commelina diffusa*, *Bulbine frutescens*, *Elephantorrhiza elephantina*, *Harpagophytum procumbens*, *Hypoxis hemerocallidea*, *Solanum lichtensteinii*, *Siphonochilus aethiopicus* and *Solanum campylacanthum* (Table 3).

In terms of the UV, the documented medicinal plants had values that ranged from 0.01–0.75 (Table 3). The most used medicinal plant was *A. elongatum*, with a UV of 0.75. The plant was mentioned by 76 knowledge holders as treatment for diverse health conditions such as umbilical cord, muscle fits, measles, bladder inflammation, weight loss and appetite. As another example of a well-utilised plant, *C. diffusa* had a UV of 0.44 with indication as an herbal remedy to treat and manage umbilical cord, purgative in children, while *E. prostrata* (UV = 0.31) was used to treat stomachaches, fever, and abdominal cramps. *Hypoxis hemerocallidea* (UV = 0.19) was used for sunken fontanelle, bladder inflammation, kidney failure, urinary tract infection, bronchitis pneumonia, child cleanse influenza and ulcer, gastro-intestinal and appetite. In addition, *E. elephantina* (UV = 0.17) was used to treat infective eczema, diarrhoea, ulcers, burns, and measles. These findings demonstrate the extensive use of these aforementioned plants in treating various ailments by local inhabitants/healers and the consciousness of indigenous peoples, which makes such medicinal plants the first choice to treat disease [63]. Most THPs undergo common training practices, hence the popularity of the plants among local traditional health practitioners as a treatment preference. Many of the plants with high UV are generally used to treat different diseases in different communities [64–66]. Asong et al. [45] reported the use of various plants from North West Province, which included *A. elongatum*, *E. prostrata*, *H. hemerocallidea* and *E. elephantina*. For instance, the whole plant of *A. elongatum* is used to treat chickenpox and yaws [45]. *Commelina diffusa* is prescribed for treating urinary tract infections, swellings, inflammation, diarrhoea, hemorrhoids, enteritis, eye irritation, conjunctivitis, and ophthalmia [67]. It is also used to treat

**Table 3. Ethno-botanical information on plants used for the treatment and management of childhood diseases and general well-being in Ngaka Modiri Molema and Bojanala districts of North West Province, South Africa.**

| Scientific name and Family [62] | Local name | Plant part and method of preparation | Childhood diseases/conditions | Administration and Dosage | [a] Plant form | [b]N | FC | UV | Cs |
|---|---|---|---|---|---|---|---|---|---|
| *Acacia caffra* (Thunb.) Willd. **Fabaceae [TPN 061]** | Poo tshetlha | Whole plant, maceration | Ulcer, sores and stop vomiting | Orally (2x/day) | H | 5 | 4.9 | 0.04 | LC |
| *Acrotome inflata* Benth. **Lamiaceae [TPN 015]** | Mogato | Whole plant Burn, maceration | Weakness, urinary tract infection, measles and sunken fontanelle | Topical, orally (3x/day) | H | 6 | 5.9 | 0.05 | LC |
| *Aloe arborescens* Mill. **Xanthorrhoeaceae [TPN 060]** | Lekgala | Leaves, Paste | Skin itching and irritation | Paste, (As needed) | S | 1 | 0.9 | 0.01 | LC |
| *Aloe maculata* All. **Xanthorrhoeaceae [TPN 010]** | Lekgala la thaba | Leaves, stem and rhizome Maceration or infusion | Bladder inflammation, urinary tract infection, umbilical cord, diarrhoea and burns | Orally and topical, (3x/day) | S | 10 | 9.9 | 0.09 | LC |
| *Aptosinum elongatum* Eng. **Scrophulariaceae [TPN 016]** | Ditantanyane | Stem, Infusion | Umbilical cord, muscle fits, measles, bladder inflammation, weight and appetite | Orally (3x/day) | H | 76 | 75.2 | 0.75 | LC |
| *Artemisia afra* Jacq. Ex Willd. **Asteraceae [TPN 059]** | Lengana | Whole plant, Decoction | Influenza | Orally (2x/day) | H | 5 | 4.9 | 0.04 | LC |
| *Asparagus exuvialis* Burch. **Asparagaceae [TPN 018]** | Tlhokabotswaro | Leaves, Decoction or maceration | Teething | Orally (2x/day) | H | 2 | 1.9 | 0.01 | LC |
| *Barleria macrostegia* Nees. **Acanthaceae [TPN 032]** | Thotsethunya | Rhizome, Poultice | Sunken fontanelle | Bathing (2x/day) | H | 1 | 0.9 | 0.01 | LC |
| *Argemone Ochroleuca* subsp. *stenopetala* (Rose) Ownbey **Papaveraceace [TPN 056]** | Sepodise | Roots, Decoction | Teething | Orally, (2x/day) | H | 1 | 0.9 | 0.01 | IA |
| *Baillonella toxisperma* Pierre. **Sapotaceae [TPN 030]** | Mpumbulo | Roots, Decoction | Sunken fontanelle | Orally, (2x/day) | S | 1 | 0.9 | 0.01 | IA |
| *Boophone disticha* (L.f.) Herb. **Amaryllidaceae [TPN 054]** | Lesoma | Rhizome/bulb, Infusion | Ringworm | Topical, 3x/day | H | 2 | 1.9 | 0.01 | LC |
| *Boscia cf. foetida* Schinz subsp. *minima* Toelken. **Capparaceae [TPN 020]** | Motsetsigaralele | Leaves, Decoction | Sunken fontanelle and weaning | Orally, (3x/day) | H | 3 | 2.9 | 0.02 | LC |
| *Bulbine frutescens* (L) Willd. **Xanthorrhoeaceae [TPN 004]** | Makgabenyane | Rhizome/bulb, roots Infusion, maceration | Sunken fontanelle, umbilical cord; body rash, sores, phlegm and urinary tract infection | Topical and orally (2x/day) | H | 22 | 21.7 | 0.2 | LC |
| *Cadaba aphylla* (Thunb.) Wild. **Capparaceae [TPN 052]** | Monna-montsho | Leaves, roots decoction, maceration | Cleansing the child, sunken fontanelle | Orally (2x/day | H | 5 | 4.9 | 0.04 | LC |
| *Centaurea scabiosa* L. **Asteraceae [TPN 055]** | Cornflower | Whole plant, infusion | Wounds and sunken fontanelle | Topical, (1x/day) | H | 3 | 2.9 | 0.02 | IA |
| *Combretum apiculatum* Sond. subsp *apiculatum* **Combretaceae [TPN 058]** | Kgosi ya ditlhare | Rhizome/bulb, maceration or poultice | Infective eczema | Orally, (1x/day) | H | 1 | 0.9 | 0.01 | LC |
| *Combretum hereroense* var. *parvifolium* (Engl.) Wickens Kew Bull **Combretaceae [TPN 005]** | Makakaba | Leaves, Infusion | Constipation | Enema (Once) | H | 2 | 1.9 | 0.01 | LC |
| *Commelina diffusa* Burm.f. **Commelinaceae [TPN 039]** | Kgopokgolo | Rhizome/bulb, Decoction | Umbilical cord, purgative the child, preventing evil spirits and weak child | Orally, (2x/day) | H | 45 | 44.5 | 0.44 | LC |

(*Continued*)

**Table 3.** (Continued)

| Scientific name and Family [62] | Local name | Plant part and method of preparation | Childhood diseases/conditions | Administration and Dosage | [a] Plant form | [b]N | FC | UV | Cs |
|---|---|---|---|---|---|---|---|---|---|
| *Corchorus olitorius* f. *grandifolius* De Wild. **Malvaceae** [TPN 048] | Juta mellow | Roots, decoction | Constipation and cramp | Orally (2x/day) | H | 2 | 1.9 | 0.01 | LC |
| *Cotyledon orbiculata* L. **Crassulaceae** [TPN 04] | Tsebe ya kolobe | Stem, maceration | Pain, inflammation, sunken fontanelle and constipation | Paste, Topical and orally (3x/day) | H | 6 | 5.9 | 0.05 | LC |
| *Cullen tomentosum* (Thunb) J.W.Grimes. **Fabaceae** [TPN 043] | Mojakubu | Whole plant, maceration | Rash and sores | Topical, orally (3x/day) | H | 6 | 5.9 | 0.05 | LC |
| *Clutia pulchella* var. *obtusata* (Sond.) Müll.Arg. **Peraceae** [TPN 007] | Pudimolwetsi | Leaves, maceration | Weaning | Orally (3x/day) | H | 2 | 1.9 | 0.01 | LC |
| *Dianthus mooiensis* F.N Williams subsp. *Kirkii* (Burtt Davys) SS Hooper. **Caryophyllaceae** [TPN 053] | Letlhoka la tsela | Roots, whole plant | Weaning, sunken fontanelle and body rash | Orally and topical (2x/day) | H | 7 | 6.9 | 0.06 | NE |
| *Dicerocaryum senecioides* (Klotzsch) Abels. **Pedaliaceae** [TPN 011] | Tshetlho ya mibitla e mebedi | Thorn, decoction or maceration | Body rash | Topical and orally (3x/day) | H | 5 | 4.9 | 0.04 | LC |
| *Dichrostachys cinerea* (L.) Wight & Arn. **Fabaceae** [TPN 051] | Moselesele | Leaves, infusion | Diarrhoea | Orally, (As needed) | T | 1 | 0.9 | 0.009 | LC |
| *Dicoma anomala* Sond. **Asteraceae** [TPN 029] | Tlhonya | Whole plant, decoction or poultice | Diarrhoea, body rash and sunken fontanelle | Orally and topical (3x/day) | H | 9 | 8.9 | 0.08 | LC |
| *Disparago anomala* Schltr. Ex Levyns. **Asteraceae** [TPN 041] | Mojakabomo | Rhizomes, whole plant maceration and decoction | Sunken fontanelle, weaning and constipation | Topical (2x/day) Orally (3x/day) | H | 6 | 5.9 | 0.05 | LC |
| *Elephantorrhiza elephantina* (Burch) Skeels. **Fabaceae** [TPN 051] | Mositsane | Roots, maceration or poultice | Infective eczema, diarrhoea, ulcer, burns and measles | Orally and topical (3x/day) | H | 18 | 17.8 | 0.17 | LC |
| *Eucomis autumnalis* (Mill). Chitt **Asparagaceae** [TPN 052] | Mathubadifala | Roots, infusion | Urinary inflammation, oral blisters and infective eczema | Orally and topical (2x/day) | H | 7 | 6.9 | 0.06 | LC |
| *Euphorbia prostrata* Aiton. **Asparagaceae** [TPN 019] | Letswetlane | Rhizome, enema or decoction | Constipation and phlegm | Orally, (As needed) | H | 32 | 31.6 | 0.31 | NE |
| *Euphorbia serpen* Kunths. **Euphorbiaceae** [TPN 030] | Lwetsane | Rhizome/bulb, maceration | Weaning and sunken fontanelle | Orally (1x/day) and topical (As needed) | H | 3 | 2.9 | 0.02 | LC |
| *Eucalyptus camaldulensis* Dehnh. **Myrtaceae** [TPN 047] | Eucalyptus | Leaves and bark, whole plant maceration and decoction | Ringworm, tuberculosis and influenza | Orally (3x/day) | T | 6 | 5.9 | 0.05 | LC |
| *Gomphocarpus fruticosus* (L.) W.T.Aiton. **Apocynaceae** [TPN 012] | Segamelamatshi | Whole plant, poultice | Sunken fontanelle | Topical, orally and enema (2x/day) | S | 1 | 0.9 | 0.009 | LC |
| *Harpagophytum procumbens* (Burch.) DC ex Meisn. subsp. **Pedaliaceae** [TPN 033] | Sengaparile | Whole plant, decoction | Bladder inflammation, kidney failure, pneumonia, liver failure, pain and inflammation, urinary tract infection, gaining weight and bronchitis | Orally, (3x/day) | H | 12 | 11.8 | 0.11 | NE |
| *Helichrysum nudifolium* (L). Less. **Asteraceae** [TPN 40] | Motlhatlhabadimo/ Mphepho | Whole plant, burning | Enhance child growth | Paste, (As needed) | H | 2 | 1.9 | 0.01 | LC |

*(Continued)*

**Table 3.** (Continued)

| Scientific name and Family [62] | Local name | Plant part and method of preparation | Childhood diseases/conditions | Administration and Dosage | [a] Plant form | [b]N | FC | UV | Cs |
|---|---|---|---|---|---|---|---|---|---|
| *Helichrysum paronychioides* DC. Humbert **Asteraceae[TPN 037]** | Phate ya ngaka | Roots, whole plant, leaves decoction, poultice | Sunken fontanelle, ulcer, bladder inflammation, influenza and nappy rash | Topical and orally (2x/day) | H | 13 | 12.8 | 0.12 | LC |
| *Hibiscus calyphyllus* Cav. **Malvaceae [TPN 017]** | Motshididi | Roots, maceration | Impetigo | Orally, (2x/day) | H | 2 | 1.9 | 0.01 | LC |
| *Hilliardiella elaeagnoides* (DC) Swelank & J.C. Manning. **Asteraceae [TPN 025]** | Ntshikologa | Whole plant decoction | Diarrhoea | Orally (3x/day) | H | 1 | 0.9 | 0.009 | LC |
| *Hypoxis hemerocallidea* Fisch., C.A.Mey. & Ave-Lall. **Hypoxidaceae [TPN 058]** | Tshuka ya poo | Roots, decoction | Sunken fontanelle, bladder inflammation, kidney failure, urinary tract infection, bronchitis pneumonia, child cleanse influenza and ulcer, gastro-intestinal and appetite | Orally, Topical and orally (2x/day) | H | 20 | 19.8 | 0.19 | LC |
| *Hydnora africana* Thunb. **Hydnoraceae [TPN 031]** | Letlhoele | Rhizome/bulb, maceration | Coughing blood | Orally, (As needed) | H | 2 | 1.9 | 0.01 | LC |
| *Ipomoea oblongata* E.Mey.ex Chiosy A. **Convolvulaceae [TPN 001]** | Mokatelo/morobe | Rhizomes, Poultice and maceration | Chickenpox, measles and umbilical cord and appetite | Topical and orally (2x/day) | H | 12 | 11.8 | 0.11 | LC |
| *Lycium horridum* Thunb. **Solanaceae [TPN 042]** | Motlhalawadikonyana | Roots, leaves burns, infusion | Umbilical cord, warts, skin irritation and sunken fontanelle | Topical, (3x/day) | S | 3 | 2.9 | 0.02 | LC |
| *Nigella sativa* L. **Ranunculaceae [TPN 035]** | Blackseed | Whole plant, poultice | Rebuilding cells Burns | Orally and topical, As needed | H | 3 | 2.9 | 0.02 | IA |
| *Opuntia ficus-indica* (L) Mill. **Cactaceae [TPN 022]** | Toorofeye | Leaves, maceration | Diabetes and cholesterol | Orally, (3x/day) | S | 4 | 3.9 | 0.03 | LC |
| *Ozoroa paniculosa* (Sond.) R. Fern. & A.Fern. **Anacardiaceae [TPN 034]** | Monokwana | Bark, maceration | Impetigo and body rash | Topical (2x/day) | H | 4 | 3.9 | 0.03 | LC |
| *Pentanisia prunelloides* (Klotzsch ex Eckl. & Zeyh.) Walp. **Rubiaceae [TPN 027]** | Setimamollo | Roots, burning, decoction, maceration or poultice | Cancer, impetigo, skin irritation, sunken fontanelle and chicken pox and oral blisters | Topical and orally or paste (3x/day) | H | 3 | 2.9 | 0.02 | LC |
| *Punica granatum* L. **Lythraceae [TPN 021]** | Pamagrante | Bark, decoction | Diarrhoea | Orally, (As needed) | T | 10 | 9.9 | 0.09 | NE |
| *Prunus persica* (L.) Batsch. **Rosaceae [TPN 013]** | Perekisi | Leaves, decoction | Constipation and cramp | Orally, (As needed) | T | 1 | 0.9 | 0.009 | LC |
| *Ricinus communis* L. **Euphorbiaceae[TPN 010]** | Mokura | Leaves, maceration | Chicken pox | Topical, (3x/day) | S | 2 | 1.9 | 0.01 | NE |
| *Sansevieria hyacinthoides* (L) Druce. **Asparagaceae [TPN 014]** | Mosekelatsebeng | Leaves, poultice | Earache and sores | Topical (As needed) | H | 4 | 3.9 | 0.03 | LC |
| *Searsia pyroides* (Burch.) Moffett. **Anacardiaceae [TPN 038]** | Mohitla/ Bohitlha | Leaves, decoction | Influenza | Orally, 2x/day | T | 2 | 1.9 | 0.01 | LC |
| *Senna italica* subsp. arachoides Burch Lock. **Fabaceae [TPN 028]** | Sebetebete | Roots, maceration or decoction | Constipation and ulcer | Orally, (2X/day) | H | 4 | 3.9 | 0.03 | LC |
| *Senna tora* (L.) Roxb. **Fabaceae [TPN 009]** | Monepenepe | Roots, decoction | Diarrhoea | Orally, (3x/day) | S | 1 | 0.9 | 0.01 | LC |

(*Continued*)

**Table 3.** (Continued)

| Scientific name and Family [62] | Local name | Plant part and method of preparation | Childhood diseases/conditions | Administration and Dosage | ᵃ Plant form | ᵇN | FC | UV | Cs |
|---|---|---|---|---|---|---|---|---|---|
| *Siphonochilus aethiopicus* (Schweinf.) B.L.Burtt. **Zingiberaceae** | Serokolo | Rhizome, decoction | Weaning, sunken fontanelle, influenza, appetite, ulcer and diarrhoea | Topical and orally (3x/day) | H | 15 | 14.8 | 0.14 | CE |
| *Solanum campylacanthum* Hochst. ex A.Rich. subsp. *panduriforme* (Drege ex Dunal) J. **Solanaceae** [TPN 002] | Tolwane nnye | Roots, Maceration and decoction | Sunken fontanelle, umbilical cord, bladder inflammation and gastroenteritis | Orally (2x/day) | H | 14 | 13.8 | 0.13 | LC |
| *Solanum lichtensteinii* Willd. **Solanaceae** [TPN 036] | Tolwane (Kgaba) | Roots, flower Maceration and decoction | Sunken fontanelle, bladder inflammation, umbilical cord, stop vomiting and enhance growth in children | Topical and orally (3x/day) | H | 18 | 17.8 | 0.17 | LC |
| *Sutherlandia frutescens* (L.) R.Br. **Fabaceae** [TPN 006] | Lerumolamadi | Whole plant, Decoction | Body rash, bladder inflammation, kidney failure and Urinary tract infection | Orally (3x/day) | S | 5 | 4.9 | 0.04 | NE |
| *Teucrium sessiliflorum* Benth. **Lamiaceae** [TPN 026] | Setlhokotlhko | Leaves, Maceration | Diarrhoea and sunken fontanelle | Orally and topical (3x/day) | H | 5 | 4.9 | 0.04 | NE |
| *Warburgia salutaris* (Bertol. f.) Chiov. **Canellaceae** [TPN 049] | lekwati/Molaka | Leaves, stem and bark, Maceration and decoction | Pneumonia, influenza, sores | Steaming Orally (2x/day) | T | 4 | 3.9 | 0.03 | EN |
| *Withania somnifera* (L) Dunal. **Solanaceae** [TPN 003] | Modikasope | Roots, leaves Maceration | Sunken fontanelle, restless and weaning, for sores or pulse and constipation | Topical, orally (3x/day) | S | 11 | 10.8 | 0.10 | LC |
| *Ziziphus oxyphylla* Edgew. **Syn:** *Ziziphus acuminata* Royle. **Rhamnaceae** [TPN 024] | Sekgalofatshe | Leaves and bark, Poultice | Diarrhoea | Orally, (2x/day) | T | 6 | 5.9 | 0.05 | LC |

The botanical names of the plants were verified using the World flora online (http://www.worldfloraonline.org/), and conservation status were verified using the South African Red data list (http://redlist.sanbi.org/species)

ᵃPlant form: T = Tree, S = Shrub and H = Herb.

ᵇN = Number of participants. Ethnobotanical Index used, N = Frequency of Citation; Use-value = UV; FC = Frequency of Citation; Conservation status = Cs;

CE = Critically Endangered; NE = Not Evaluated; LC = Least common; IA = Invasive alien species and EN = Endangered. The conservation status were verified using the South African Red data list (http://redlist.sanbi.org/species)

jaundice in children in the Windward Islands and Cuba [68]. On the other hand, *E. prostrata*, has been used to treat skin problems, migraines, intestinal parasites, and warts as a medicinal herb [69]. *Hypoxis hemerocallidea* also known as 'African potato', is widely used in South African traditional medicine to cure, manage various health conditions including childhood convulsions and epilepsy [70]. In the study by Mhlongo et al. [71], *H. hemerocallidea* was identified among the most cited plants with the highest priority. *Elephantorrhiza elephantina* is used for gastrointestinal diseases, respiratory ailments, pain and inflammation [72].

Factors such as availability and indigenous success may influence the popular use of these medicinal plants within the cultural folklore and primary health care amongst the local communities [73, 74]. Furthermore, Phillips et al. [40] hypothesized that a high UV indicates the cultural diversity attached to the plants. Though unknown to the knowledge holders, there is the possible presence of biologically active compounds in these plants that could be responsible for the perceived healing effects, which may be worth investigating. Notably, plants with lower

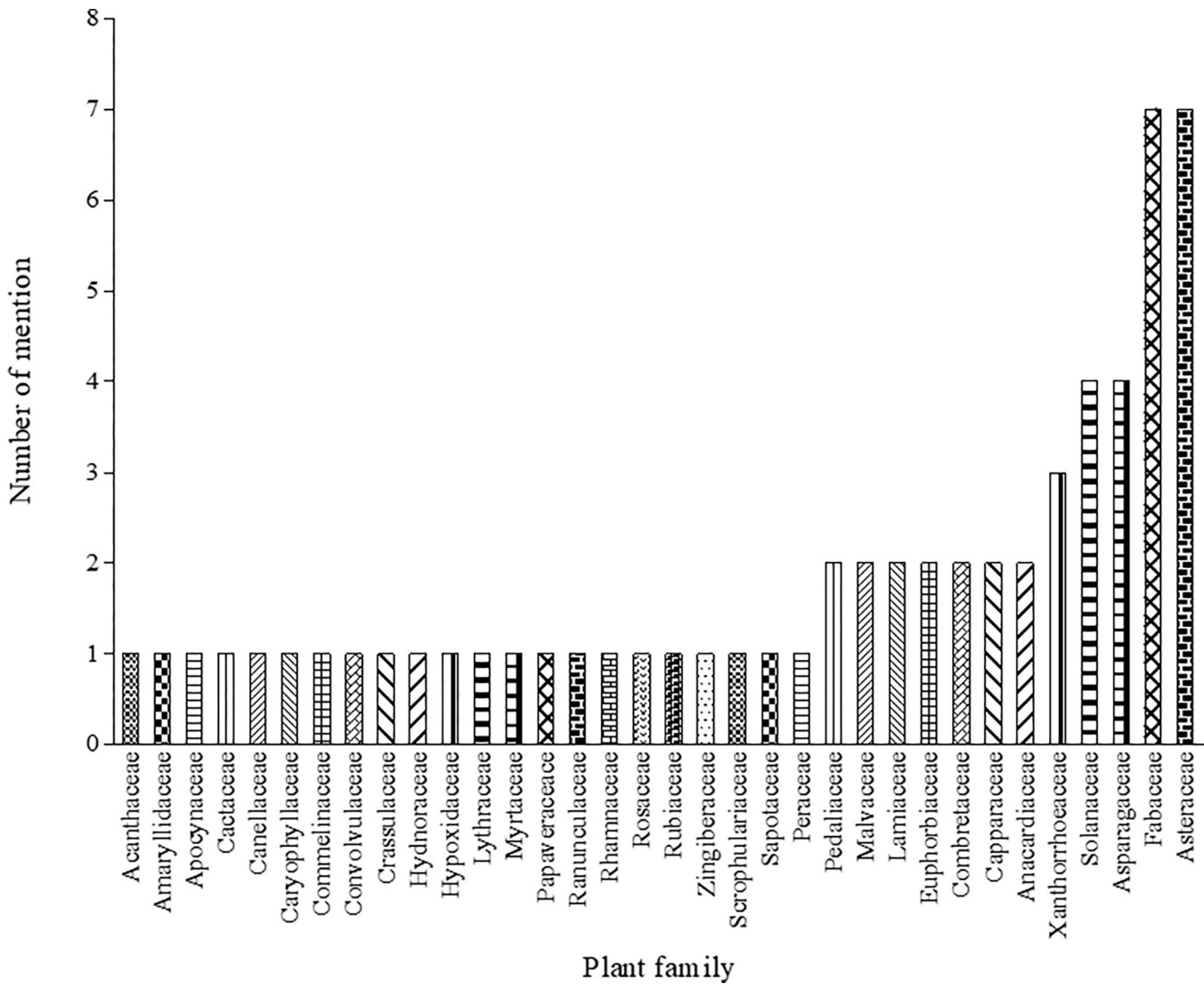

**Fig 2. Frequency of plant families used to manage and treat childhood diseases in Ngaka Modiri Molema and Bojanala districts of North West Province, South Africa.**

UV, such as *Prunus persica* (UV = 0.9) and *Aloe arborescens* (UV = 0.01), should not be ignored because local healers indicated that these plants exert potent effects [75]. According to some knowledge holders, some plants with low UV were in high use in the past. However, the continuous destruction of habitats resulting in the scarcity necessitate the search for alternative plants.

## 3.4 Informant census factors

The categories of childhood diseases and informant consensus factors (ICF) are shown in Table 4. A total of 29 different childhood diseases are being treated and managed with 61 medicinal plants documented in the study area. The seven ailment categories identified were pain and inflammation diseases, gastro-intestinal related diseases, oral-related diseases, urinary-related diseases, skin-related diseases, respiratory-related diseases, and social/nutritional-

**Table 4. Childhood diseases based on categories and informant consensus factor (ICF) in Ngaka Modiri Molema and Bojanala districts of North West Province, South Africa.**

| Category | No of use reports (Nur) | No of plant taxa (Nt) | ICF |
|---|---|---|---|
| **Gastro-intestinal diseases** | **242** | **18** | **0.92** |
| Diarrhoea | | | |
| Constipation | | | |
| Gastroenteritis | | | |
| **Pain and inflammation** | **127** | **10** | **0.9** |
| Umbilical cord | | | |
| Pain and inflammation | | | |
| Earache | | | |
| **Urinary genital diseases** | **190** | **17** | **0.9** |
| Bladder inflammation | | | |
| Kidney failure | | | |
| Urinary tract infection | | | |
| **Oral-related diseases** | **61** | **7** | **0.9** |
| Oral blisters | | | |
| Phlegm | | | |
| Stop vomiting | | | |
| Teething | | | |
| **Skin-related diseases** | **381** | **34** | **0.99** |
| Body rash | | | |
| Skin irritation | | | |
| Sores | | | |
| Impetigo | | | |
| Infective eczema | | | |
| Measles | | | |
| Chicken pox | | | |
| Ringworm and warts | | | |
| **Respiratory diseases** | **42** | **15** | **0.62** |
| Pneumonia | | | |
| Bronchitis | | | |
| Influenza | | | |
| Tuberculosis | | | |
| **Social/nutritional health conditions** | **90** | **34** | **0.65** |
| Sunken fontanelle | | | |
| Restless and weaning | | | |
| Appetite, enhance child growth and gain weight | | | |
| Sugar diabetes and cholesterol | | | |

related diseases. The ICF of different categories of childhood diseases ranged from 0.62–0.99. Skin-related diseases had the highest ICF value (0.99). This category had 381 use-reports, comprising 34 plants (55% of total plants) used for childhood diseases (Table 4). The most cited plants were *B. frutescens* and *E. elephantina*. This indicates a high consensus on the use reports of these medicinal plants. Asong et al. [45] reported using these two plants for skin-related diseases with a high UV.

The second highest ICF (0.92) was linked to gastrointestinal-related diseases. This category had 242 uses reports and three gastrointestinal-related diseases; with diarrhoea being the most

cited. Diarrhoea is the most common clinical manifestation of gastrointestinal diseases and can be caused by both infectious and non-infectious agents [76]. This disease may be abrupt and self-limiting in immune-competent individuals, especially in people with underlying debilitating clinical conditions such as HIV/AIDS and diabetes mellitus. Stats SA [28] indicated that the North West Province suffers from inadequate and lack of access to clean and safe water, as a result this maybe one of the major vehicles for transmission of gastrointestinal-related diseases. The most cited plant in this category was *E. elephantina*. There was a wide distribution of the plant taxa in this category. Moreover, the high ICF value indicates an agreement among the participants regarding the plant taxa used for treatment and possibly, a high prevalence of diseases in this category [77].

Respiratory-related diseases category recorded the lowest ICF value (Table 4), indicating that there was less knowledge exchange amongst the participants on medicinal plants used to treat and manage this category of diseases. This category had 47 use reports, 25 plant species and 3 respiratory-related diseases. This could be influenced by other respiratory-related diseases such as influenza have known remedies that are used to treat and manage this type of disease is common knowledge and are also based on individuals' beliefs. According to Gazzaneo et al. [42], ICF values range from 0 to1 with values closer to 0, meaning that there is a lesser degree of agreement among the participants on the plants used to treat and manage diseases in a particular category. The lower ICF in this category indicates a lower degree of agreement among the participants regarding the medicinal plants used in the treatment and management of respiratory-related diseases in the study area. These low ICF values recorded in the present study could be ascribed to the recent trends in emerging diseases, including Covid-19 [78]. Besides, the low ICF values for respiratory-related diseases could be explained by the fact that these diseases were not essential health problems at that time of the data collection. However, these types of diseases, mainly skin-related disease (Infective eczema) and gastrointestinal (constipation, diarrhoea), are commonly referred to as THPs and are generally treated and managed with polyherbal medicines; thus, a range of medicinal plants was reported (Table 4). Furthermore, this could be that socio-economic status of living and poor social facilities are poor states in the North West Province [27, 28].

### 3.5 Plant parts used for childhood diseases

In the current study, the participants harvested different plant parts to prepare traditional remedies (Fig 3). Roots and rhizomes were the most frequently used as botanical drugs (40%), followed by leaves (23%) and whole plants (20%). The dominance of underground parts observed in this study resonates with the findings of several studies in, South Africa [72, 79, 80] and other countries [15, 81]. Moichwanetse et al. [52] articulated that Batswana people believe in "ditswammung," which means "out of the soil". This is because it is believed that underground parts contain higher concentrations of the active ingredients and are storage organs of secondary metabolites [82, 83]. However, harvesting parts such as bulbs, rhizomes, and bark increase the conservational strains on the ecosystem, which often exacerbate the sustainability of medicinal plants mainly when extensively harvested in the wild [84, 85]. To ensure sustainable utilization of medicinal plant resources, it is necessary to apply proper harvesting strategies and conservation measures. This can be facilitated through increasing public awareness and promoting the cultivation of medicinal plants [86].

### 3.6 Preparation, administration, and dosage methods

Participants reported a variety of herbal preparations used to prepare traditional medicine for different types of childhood diseases in the study area. Maceration (39%), decoction (38%) are

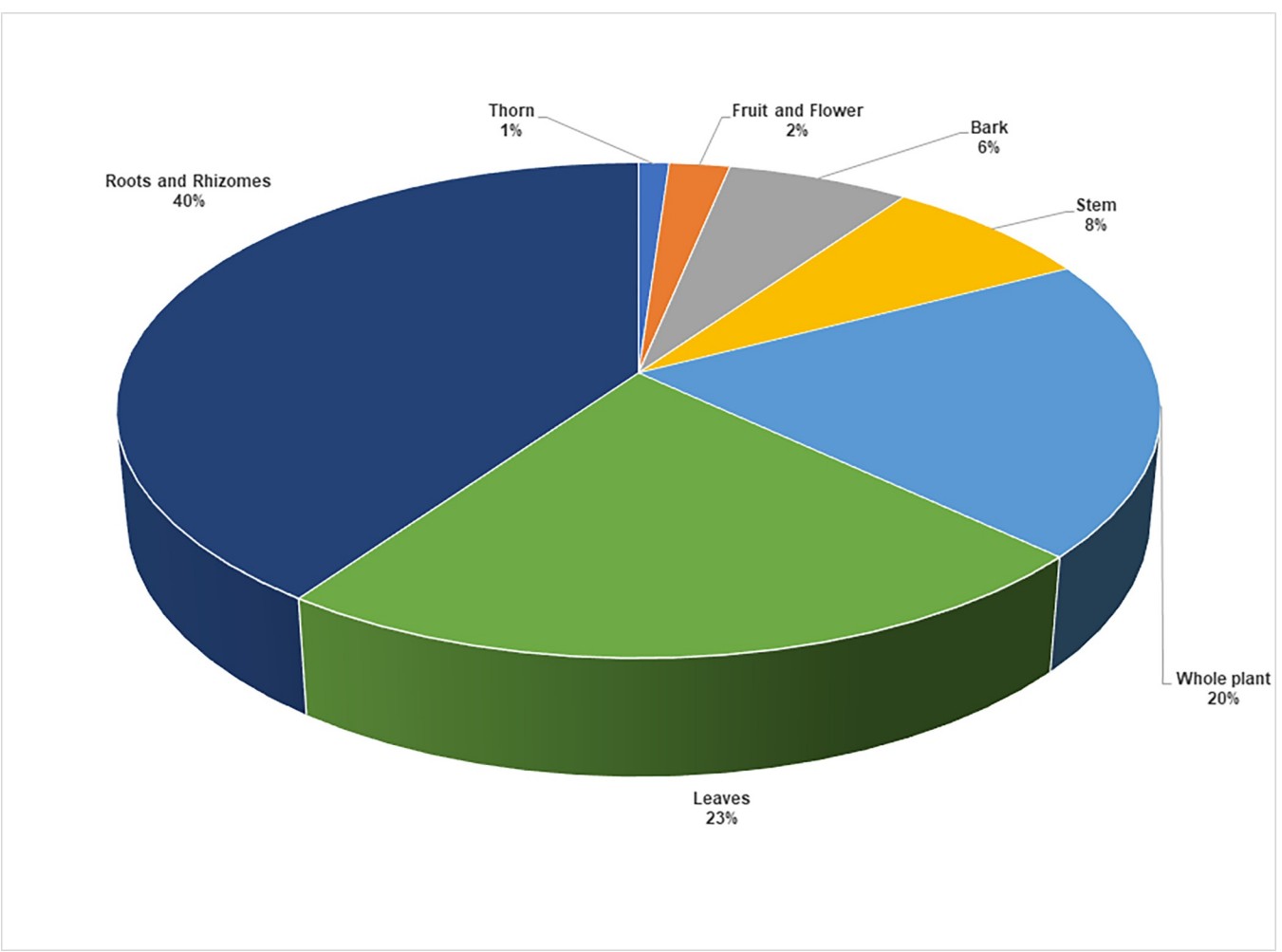

**Fig 3. Distribution (%) of medicinal plant parts used to manage and treat childhood diseases in Ngaka Modiri Molema and Bojanala districts of North West Province, South Africa (n = 92).**

the most utilised forms of preparation, followed by poultice (14%) and infusion (9%), which is used for the least number of plant species. Application of the medicinal plant preparations is done through three different routes of administration. Oral application (60%) was the most frequently cited route of administration, followed by topical (39%), while applications through enema was generally low (1%). Medicinal plants used for skin-related diseases were prepared using maceration and applied topically as lotion. Plants such as *A. arborescens*, *E. elephantina*, *H. paronychioides*, and *H. hemerocallidea* were popular species used for skin-related diseases. Furthermore, decoction and oral administration were the second most popular methods of preparation and administration, primarily in plants such as *E. prostrata*, *D. anomala*, *P. granatum*, and *Z. oxyphylla*, reported as a treatment for gastro-intestinal diseases (diarrhoea and constipation). Similar results were also observed in the previous study, where plants such as *A. arborescens*, *E. elephantina*, and *H. hemerocallidea* were applied directly to skin-related diseases [45]. Maceration and decoction are the most common modes of preparation in the current study and correlates with a few previous findings [87, 88]. The largest proportion of remedies prepared as a powder in the present study is not in concordance with some other research in countries such as Uganda [51] and Kenya [89].

### 3.7 Life-form of the identified plants

In the current study, the life-form for the documented plants included herbs, shrubs and trees with a distribution of 68, 17 and 15%, respectively. This does not reflect the floristic composition of the vegetation of the North West Province and the weather conditions experienced in the region [26, 90]. Most regions in the North West Province fall within the savannah biome with its associated Bushveld vegetation, and the western region primarily comprises Kalahari thornveld and shrub bushveld. In contrast, the central region is dominated by dry *Cymbopogon-Themeda* veld [90–92]. The high usage of herbs in treating and managing childhood diseases is a promising finding from a conservation perspective given that herbs re-generate and grow faster after being harvested. In North West Province, 40% of the ecosystems are under severe stress, with 11 of the 61 vegetation and 14 of the 18 river types classified as threatened in terms of ecosystem status [93].

### 3.8. Commonly encountered childhood diseases in the study areas

The nominal group technique and the interviews enabled the researcher to identify 7 disease categories generated from the prevalent 29 sub-diseases as classified by the participants (Table 4). The most prevalent categories of childhood diseases were skin-related diseases (burns, skin irritation and warts). Skin-related diseases are still a huge concern and cause morbidity among children in developing countries, especially in sub-Saharan Africa [94, 95]. The prevalence of skin-related diseases in children under <5 years is well recognised [96, 97]. According to Hay et al. [98], skin infections contribute to approximately 34% of occupational health diseases globally. Notably, the high occurrence of skin-related diseases in children could be associated with their low immune systems or low socio-economic status, favorable tropical weather, neglect and poor hygienic living conditions, including the lack of clean water and sanitation, particularly in the remote areas [95]. The high prevalence of skin-related diseases observed in the current study may be partially associated with opportunistic skin infections and HIV/AIDS, which are usually the first signs of HIV infection [77]. North West Province has high inactive cases of HIV/AIDs [99], which are often exacerbated by overcrowding and unhygienic environments [25]. The current findings differ from those of Boschi-Pinto et al. [100], whereby gastro-intestinal disease such as diarrhoea was a major cause of morbidity and mortality among children aged 5 in Sub-Saharan Africa. In 2008, among the estimated mortality of 4.2 million children who are 5-year-old in Africa, diarrhoea caused the largest proportion (19%), followed by pneumonia and malaria. Previous studies indicated that co-morbidity, poor nutritional status, dehydration, lack of breastfeeding and prolonged diarrheal duration were risk factors for death in young children in Africa [72, 101].

In this study, gastro-intestinal diseases were the second most prevalent diseases. Gastrointestinal diseases affect the gastrointestinal tract from the mouth to the anus, including nausea/vomiting, food poisoning, lactose intolerance and diarrhoea [102]. According to the World Health Organization (WHO) [95], hygiene may be the leading cause of children-related diseases. Diarrhoea is the most common paediatric sickness and one of the leading causes of infant and child mortality [103, 104]. Diarrhoea morbidity and mortality are prevalent in developing countries, especially in Africa [76, 105]. According to Stats SA [25], approximately 49.2% of the population in the North West Province lives below the upper-bound poverty line with no access to proper housing, water and sanitation. The majority of children reside in formal households, 8.6% in informal dwellings, 13.3% in traditional structures and only 0.3% resided in other types of dwellings [25]. These poor conditions likely contribute to these diseases and may explain the high number of local communities resorting to traditional medicine. Similar findings were observed from previous studies in African countries such as Uganda [106] and Zimbabwe [107].

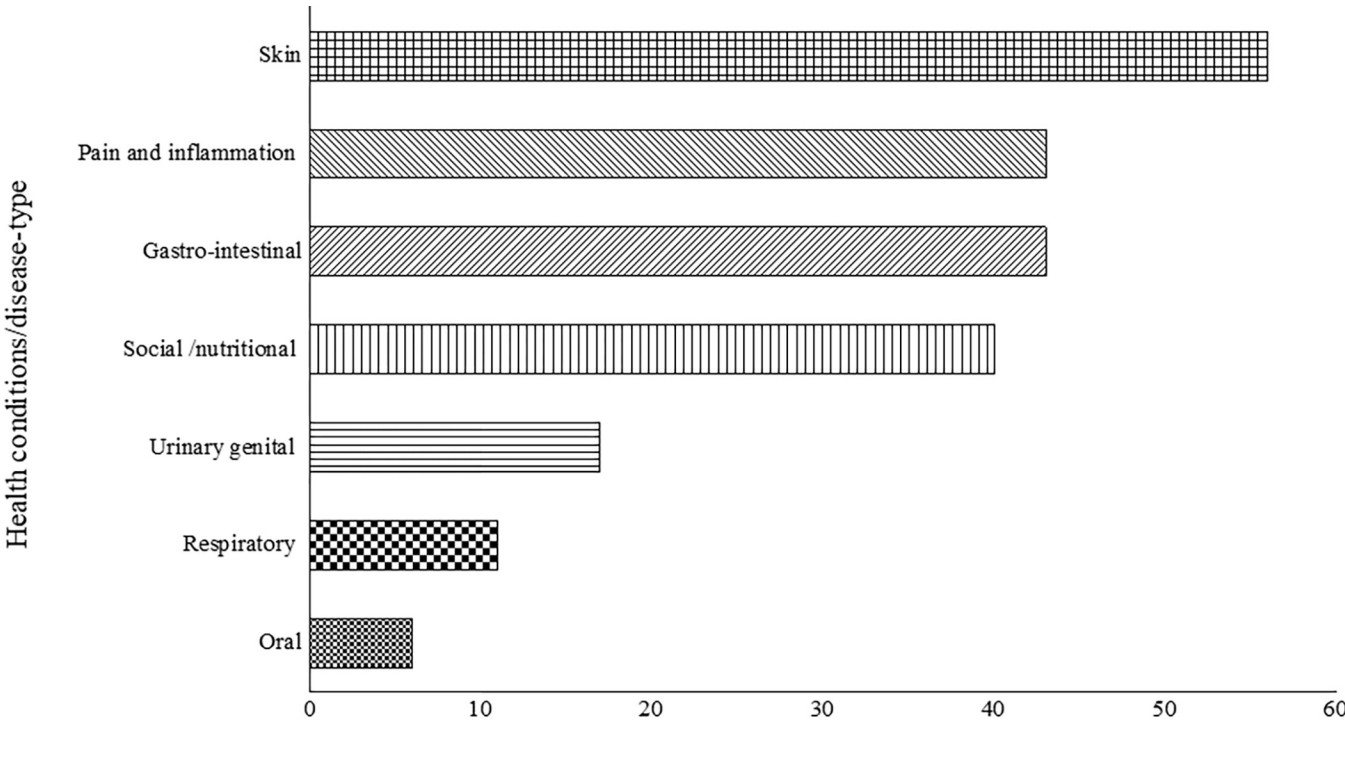

**Fig 4. The range of childhood diseases treated and managed by the participants in Ngaka Modiri Molema and Bojanala districts of North West Province, South Africa.**

Pain and inflammation diseases, including urinary, genital diseases (such as bladder inflammation, kidney failure and urinary tract infection), and the umbilical cord was among the largest group behind skin-infections and related gastrointestinal diseases (Fig 4). Amongst the pain and inflammation categories, the umbilical cord (*Lelana la mokhubu*) was the most prevalent disease in this category. About a quarter of the global neonatal deaths are due to umbilical cord infection; 75% of these occur in the first week of life, with the umbilical cord being the cause of death [108]. Most of the participants indicated that umbilical cord infections (*omphalitis*) and neonatal *sepsis* are significant contributors to the proportion of neonatal infections that prove fatal in the rural areas. These results were similar to the studies done in Nepal, Bangladesh, and Pakistan, which indicated that umbilical cord was recorded as one of the most prevalent childhood health condition [109, 110].

Other childhood-related diseases included sunken fontanelle diseases (*Tlhogwana ya bana*). It is known to contribute significantly to the mortality and morbidity of children < 5 years in low-middle-income countries [111]. In most developing countries, dehydration and sunken fontanelle are not often attributed to physical conditions. Sunken fontanelle is often associated with forces outside the body, social and spiritual. In many cultures, people believe that small children need special protection because they are very vulnerable to diarrhoea associated with a sunken fontanelle [112]. In the current study, about 90% of the participants indicated that sunken fontanelle is a complex collection of illnesses with diarrhoea as a symptom, and they did not have a concept of dehydration. Depressed fontanelle was recognized as a sign of serious illness, and diarrhoea was linked to symptoms of dehydration through the concept of "*Nogana ya bana*" which means sunken fontanel.

### 3.9 Conservation status

In terms of conservation status, the medicinal plants were categorized as "Least Concerned" (79%), "not evaluated" (11%), "Invasive Alien Species" (6%), and "Critical and Endangered species" (4%). The current finding suggests that most medicinal plants used to treat and manage childhood diseases in the study locations are readily available and not under severe conservation strains. Based on a recent review by Ndhlovu et al. [13], an estimated 84% of medicinal plants used for childhood-related diseases are listed as 'least concern' in South Africa. Both *S. aethiopicus* and *W. salutaris* are medicinal plants whose wild populations are reported to be rapidly declining and endangered. Similar results have been observed in different studies [86, 113, 114]. South Africa is globally recognised as a prolific habitat for flora and fauna [115], with at least 3000 higher plant species with therapeutic value. However, medicinal plants used to treat and manage the disease are mostly harvested from the wild [60], with the possibility of many facing extinction from uncontrolled harvesting. Several studies have indicated the significance of the conserving the flora and fauna [84, 116].

### 3.10 Multiple treatment indications and combination of medicinal plants

The participants mentioned 6 plants in the inventory with multiple indications (uses), either as single or poly-plant remedies (Table 5). Single-plant remedies with multiple indications included plants such as *A. elongatum*, *D. mooiensis*, *C. diffusa*, *E. serpen* and *S. hyacinthoides*. In the study area, these above-mentioned plants had the highest citation as poly-plant remedies for treating and managing childhood diseases (Table 5). The use of two or more plants reflects the concept of synergy, where the association of plants can result in enhanced therapeutic efficacy. Several studies indicate that the mono-substance therapy model has gradually shifted toward adopting combination therapies, in which multiple active components are employed [117, 118]. Moichwanetse et al. [52] identified five medicinal plants in North West Province that are prescribed as poly-plant remedies in ethnoveterinary. According to Li and Weng [119], traditional medicine often comprises several ingredients mixed in a given ratio as herbal formula. Each ingredient in isolation sometimes lacks therapeutic activities seen in the holistic formulation, a phenomenon known as the combinatorial effect.

## 4. Conclusions

The current findings revealed that childhood diseases are still a common problem in rural areas of the North West Province and local communities continue to rely on plants to meet the healthcare needs of their children. This is the first ethnobotanical study in North West

**Table 5. Plant combinations used for treating childhood diseases in Ngaka Modiri Molema and Bojanala districts of North West Province, South Africa.**

| Scientific name | Local name | #Plant part | Application and preparation | Type of disease |
|---|---|---|---|---|
| *Commelina diffusa* Burm.f. + *Aptosinum elongatum* Eng. | Ditantanyane + Kgopokgolo | R+B | Decoction, orally | Umbilical cord |
| *Sansevieria hyacinthoides* (L) Druce. + *Aptosinum elongatum* Eng + *Helichrysum paronychioides* DC. Humbert. | Mosekelatsebeng + Ditantanyane + Phate ya ngaka | R+ S+ B | Decoction, orally | Reduce weaning and muscular |
| *Commelina diffusa* Burm.f. + *Solanum lichtensteinii* Willd. | Kgopokgolo + Tolwane | B+R | Infusion with milk, orally | Umbilical cord |
| *Euphorbia serpen* Kunths + *Dianthus mooiensis* F.N Williams subsp. *kirkii* (Burtt Davys) SS Hooper. | Lwetsane + Letlhoka la tsela | B+R | Maceration, topical | Sunken fontanel and weaning |
| *Sansevieria hyacinthoides* (L) Druce. + *Commelina diffusa* Burm.f. | Makgabenyane + Kgopokgolo | B+R | Infusion, orally | Umbilical cord |

#Plant part, R = roots, B = bulbs, S = stem and L = leaves

Province relating to the use of medicinal plants for managing childhood diseases and wellbeing. We documented 61 medicinal plants and associated indigenous knowledge used to manage diverse diseases. Approximately 89% of medicinal plants were recorded for the first time as medicinal plants used for childhood-related diseases. In comparison with earlier ethnobotanical studies conducted in South Africa, 11% of the medicinal plants were confirmed to be used for childhood-related diseases. These plants included *A. arborescens*, *B. macrostegia*, *D. cinerea*, *E. autumnalis*, *H. hemerocallidea*, *S. lichtensteinii* and *W. somnifera*. In addition, the medicinal plants had similar uses with the current findings. However, we observed considerable differences with respect to the preparation and administration techniques. The current results highlighted some degree of novelty with regards to the diverse uses associated with medicinal plants in South Africa. We also demonstrated the importance of collecting new ethnobotanical information on well-known medicinal plants. Quantitative analysis revealed that *A. elongatum*, followed by *C. diffusa*, and *E. prostrata*, had the highest UV and were the most cited plant species. Based on ICF, skin-related diseases had the highest ICF value (0.99). This category had 381 use-reports, comprising 34 plants (55.7% of total plants) used for childhood-related diseases, with *B. frutescens* and *E. elephantina* being the most-cited plants in this category. This valuable plant inventory demonstrates a major step towards the ongoing effort of the documentation, preservation, and promotion of indigenous knowledge of the study area. Overall, these results enrich the national and globally pharmacopeia in managing the healthcare needs of children. However, further investigations into the efficacy, safety, biological activities, and phytochemical profiling of these documented plants remain pertinent. Such stringent investigations may uncover novel compounds with pharmaceutical relevance in treating childhood diseases.

## Acknowledgments

We are grateful to our participants for their willingness to be part of this study. We extend our sincere gratitude to the North West Dingaka Association, Traditional Council (Barolong Boo Rra Tshidi) for granting permission and access to conduct this study, and the North West Department of Economic Development, Environment, Conservation, and Tourism for the plant collection permit. We appreciate the support from the South African National Biodiversity Institute (SANBI), Pretoria, for assisting with plant identification and Ms Almari Van Niekerk for creating the map.

## Author Contributions

**Conceptualization:** Peter Tshepiso Ndhlovu, Abiodun Olusola Omotayo, Wilfred Otang-Mbeng, Adeyemi Oladapo Aremu.

**Formal analysis:** Peter Tshepiso Ndhlovu, John Awungnjia Asong.

**Investigation:** Peter Tshepiso Ndhlovu, John Awungnjia Asong.

**Project administration:** Abiodun Olusola Omotayo, Wilfred Otang-Mbeng, Adeyemi Oladapo Aremu.

**Resources:** Adeyemi Oladapo Aremu.

**Supervision:** Abiodun Olusola Omotayo, Wilfred Otang-Mbeng, Adeyemi Oladapo Aremu.

**Writing – original draft:** Peter Tshepiso Ndhlovu.

**Writing – review & editing:** John Awungnjia Asong, Abiodun Olusola Omotayo, Wilfred Otang-Mbeng, Adeyemi Oladapo Aremu.

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
