## [Decision Letter · Decision Letter 0]

1 Nov 2022

PONE-D-22-19573Ethnobotanical survey of medicinal plants used by indigenous knowledge holders to manage healthcare needs in childrenPLOS ONE

Dear Dr. Aremu,

Thank you for submitting your manuscript to PLOS ONE. After careful consideration, we feel that it has merit but does not fully meet PLOS ONE’s publication criteria as it currently stands. Therefore, we invite you to submit a revised version of the manuscript that addresses the points raised during the review process.

Both aims of the study should be improved as suggested by Reviewer #1. Sampling procedures should be better presented.

Minor corrections:

L125: "in" into "are" or "were".

Please write "Province" or "province", not both.

Please use a full stop to abbreviate the genus name (e.g., L277). Once fully mentioned, a genus should be abbreviated (please note e.g., L291, 293).

Please do not randomly capitalize terms, as in e.g., L308.

Please italicize "Cymbopogon-Themeda" in L394.

Please use full stops to abbreviate authority names in Table 5.

Please be consistent in using full genera names in Conclusion in L501-502. I suggest using full genera names in these lines.

Be sure to follow recommendations for figure preparation and submit the as separate files.

We look forward to receiving your revised manuscript.

Kind regards,

Branislav T. Šiler, Ph.D.

Academic Editor

PLOS ONE

Journal Requirements:

Additional Editor Comments:

Both aims of the study should be improved as suggested by Reviewer #1. Sampling procedures should be better presented.

Minor corrections:

L125: "in" into "are" or "were".

Please write "Province" or "province", not both.

Please use a full stop to abbreviate the genus name (e.g., L277). Once fully mentioned, a genus should be abbreviated (please note e.g., L291, 293).

Please do not randomly capitalize terms, as in e.g., L308.

Please italicize "Cymbopogon-Themeda" in L394.

Please use full stops to abbreviate authority names in Table 5.

Please be consistent in using full genera names in Conclusion in L501-502. I suggest using full genera names in these lines.

Be sure to follow recommendations for figure preparation and submit the as separate files.

Reviewers' comments:

Reviewer's Responses to Questions

**Comments to the Author**

1. Is the manuscript technically sound, and do the data support the conclusions?

Reviewer #1: No

Reviewer #2: Yes

Reviewer #3: Yes

2. Has the statistical analysis been performed appropriately and rigorously? 

Reviewer #1: I Don't Know

Reviewer #2: Yes

Reviewer #3: N/A

3. Have the authors made all data underlying the findings in their manuscript fully available?

Reviewer #1: Yes

Reviewer #2: Yes

Reviewer #3: Yes

4. Is the manuscript presented in an intelligible fashion and written in standard English?

Reviewer #1: No

Reviewer #2: Yes

Reviewer #3: Yes

5. Review Comments to the Author

Reviewer #1: This is a very minor regional ethnobotanical survey presenting a mere list of plant uses with some nonsense quantitative attempts.

The research does not the start from robust objectives (the first one is solely descriptive, while the second too large and not addressed actually by the paper), the methodology is not well described (esp. sampling) and no serious data analysis and data interpretation has been conducted.

Reviewer #2: 1. The study presents the results of original research. - Yes.

2. Results reported have not been published elsewhere. - Yes.

3. Experiments, statistics, and other analyses are performed to a high technical standard and are described in sufficient detail. - Yes. The method is clear and well executed

4. Conclusions are presented appropriately and are supported by the data. - Yes.

5. The article is presented in an intelligible fashion and is written in standard English. - yes.

6. The research meets all applicable standards for the ethics of experimentation and research integrity. - Yes.

7. The article adheres to appropriate reporting guidelines and community standards for data availability. - Yes.

Reviewer #3: The manuscript entitled, "Ethnobotanical survey of medicinal plants used by indigenous knowledge holders to manage healthcare needs in children" focuses on the exploration of the medicinal plants used for treating childhood diseases in some areas of South Africa. I think the topics is interesting and matched with the journal’s scope. The authors have also presented the results of original research. However, I believe that some suggestions below might improve the manuscript before acceptance :

- Abstract is well written; All key findings have been presented. However, the author is suggested to mention the most cited plant family used to treat childhood diseases (Asteraceae) as this family also appears in the keywords.

- The authors have shown very comprehensive information in the introduction section. The objective of the study should be integrated in the paragraph (not per point as presented in the MS draft).

- The author used standart ethnobotanical study methods. Please give number of each formula used in the study.

- The author should re-arrange the title of the table. Some information (i.e. abbreviations etc) should be placed below the table.

- Author has compared the results with other ethnobotanical studies. However, most of the references described the study from Africa. It might also better to compare the results with another region outside Africa. I belive it might enrich information obtained in the study.

- It might also be good to state the importance of plant medicinal conservation in the conclusion section. Even though, the results showed that medicinal plants reported in this study are not under severe conservation strains.

- The authors used up to date references.

6. PLOS authors have the option to publish the peer review history of their article (what does this mean?). If published, this will include your full peer review and any attached files.

Reviewer #1: No

Reviewer #2: No

Reviewer #3: No

---

## [Author Response · Author response to Decision Letter 0]

20 Jan 2023

RESPONSES TO COMMENTS: PONE-D-22-19573

Title: Ethnobotanical survey of medicinal plants used by indigenous knowledge holders to manage healthcare needs in children

Comment: L125: "in" into "are" or "were".

Response: line 125, “in” has been revised to “were” [see line 117; page 5]. 

Comment: Please write "Province" or "province", not both.

Response: Thanks for the suggestions; we have revised the “province” to “Province” for consistency throughout the manuscript [for example see line 216-217; page 8,].

Comment: Please use a full stop to abbreviate the genus name (e.g., L277). Once fully mentioned, a genus should be abbreviated (please note e.g., L291, 293).

Response: A full stop has been placed after the abbreviation; for example, Hypoxis hemerocallidea has been revised to H. hemerocallidea [see line 303 in page 18]. Furthermore, "Hypoxis hemerocallidea" in lines 301 was not abbreviated because it starts the sentence.

Comment: Please do not randomly capitalize terms, as in e.g., L308.

Response: We have used a sentence case, thank you for the observation (see page 19, line 301).

Comment: Please italicize "Cymbopogon-Themeda" in L394.

Response: Thank you for the comment. "Cymbopogon-Themeda" has been revised to "Cymbopogon-Themeda" [see line 404; page 23].

Comment: Please use full stops to abbreviate authority names in Table 5.

Response: Full stops have been place abbreviate authority names in Table 5. 

Comment: Please be consistent in using full genera names in Conclusion in L501-502. I suggest using full genera names in these lines.

Response: We have revised the full name of the genera in the conclusion, for example, Aloe arborescens to A. arborescens [see page 27; line 513].

Comment: Please provide additional details regarding participant consent. In the ethics statement in the Methods and online submission information, please ensure that you have specified what type you obtained (for instance, written or verbal, and if verbal, how it was documented and witnessed). If your study included minors, state whether you obtained consent from parents or guardians. If the need for consent was waived by the ethics committee, please include this information.

Response: This study focused on the knowledge holders who were over the age of 18 and were able to give consent, as the South African laws recognized a person over the age of 18 as an adult. The study involved participants such as traditional health practitioners (specifically those with expertise in diseases among children) and herbal vendors in the selected study area. Participants over 18 years old were selected because the research relies only on adults and knowledge experts. In addition, participants were selected based on their ability to speak and understand Setswana or English. According to Stats SA (2018), both languages are the two main languages on the study sites. The participants of both genders registered with the North West Dingaka Association (regulatory body in the province), whose names also appear in the provincial Traditional Health Practitioners (THPs) database, were targeted. The participants who met the eligibility criteria of the various categories of THPs as defined in the Traditional Health Practitioners Act 22 of 2007 included Traditional Doctors, Diviners (Sangomas), Traditional Surgeons, Traditional Birth Attendants (TBAs) and Herbalists (Constitution of the Republic of South Africa., 2007). 

Comment: Your ethics statement should only appear in the Methods section of your manuscript. If your ethics statement is written in any section besides the Methods, please move it to the Methods section and delete it from any other section. Please ensure that your ethics statement is included in your manuscript, as the ethics statement entered into the online submission form will not be published alongside your manuscript. 

Response: Thank you for the suggestions; we have moved the ethics approval from the declarations to the last aspects of the methods [see page 7; lines 184-204].

Comment: We note that Figure 1 in your submission contain [map/satellite] images which may be copyrighted. All PLOS content is published under the Creative Commons Attribution License (CC BY 4.0), which means that the manuscript, images, and Supporting Information files will be freely available online, and any third party is permitted to access, download, copy, distribute, and use these materials in any way, even commercially, with proper attribution. For these reasons, we cannot publish previously copyrighted maps or satellite images created using proprietary data, such as Google software (Google Maps, Street View, and Earth). For more information, see our copyright guidelines: 

http://journals.plos.org/plosone/s/licenses-and-copyright.

NASA Earth Observatory (public domain): http://earthobservatory.nasa.gov/ Landsat; http://landsat.visibleearth.nasa.gov/USGS EROS (Earth Resources Observatory and Science (EROS) Center) (public domain): http://eros.usgs.gov/# Natural Earth (public domain): 

http://www.naturalearthdata.com/

Response: Thanks for the comment, the map (Figure 1) used in this study has not yet been published or used in any study. It was drawn by the author based on his study location and so the copyright belong to the author. The map was specifically created for this publication.

Comment: Please review your reference list to ensure that it is complete and correct. If you have cited papers that have been retracted, please include the rationale for doing so in the manuscript text or remove these references and replace them with relevant current references. Any changes to the reference list should be mentioned in the rebuttal letter that accompanies your revised manuscript. If you need to cite a retracted article, indicate the article’s retracted status in the References list and also include a citation and full reference for the retraction notice.

Response: All the references have been reviewed to ensure they are complete and correct. We have checked and can confirm that none of the references have been retracted. See for e.g. 

• Noorhosseini, S.A., Fallahi, E. & Damalas, C.A. Promoting cultivation of medicinal and aromatic plants for natural resource management and livelihood enhancement in Iran. Environ Dev Sustain 22, 4007–4024 (2020). https://doi.org/10.1007/s10668-019-00368-7

• Mokwena Ke & Kachabe J (2022) Profile of mothers whose children are treated for malnutrition at a rural district hospital in the North West province, South Africa, South African Journal of Clinical Nutrition, 35:1, 17-22, DOI: 10.1080/16070658.2021.1921899

Comment: Has the statistical analysis been performed appropriately and rigorously? 

Reviewer #1: I Don't Know

Reviewer #3: N/A

Response: This study followed the ethnobotanical indices to analyze the data, which was proposed by Leonti, M., 2022. The relevance of quantitative ethnobotanical indices for ethnopharmacology and ethnobotany. Journal of Ethnopharmacology, 115008. Hence, we are sure that the analysis was done appropriately.

Comment: Is the manuscript presented in an intelligible fashion and written in standard English?

Reviewer #1

Response: Thank you for the observation, all the typographical or grammatical errors have been corrected at revised for example line 125, “in” has been revised to “were” [see line 117; page 5] and “province” to “Province” for consistency throughout the manuscript [for example see line 216-217; page 8,].

Comment: Please use a full stop to abbreviate the genus name (e.g., L277). Once fully mentioned, a genus should be abbreviated (please note e.g., L291, 293).

Response: A full stop has been placed after the abbreviation; for example, Hypoxis hemerocallidea has been revised to H. hemerocallidea [see lines 285,299, and 301 in page 18]. Furthermore, "Hypoxis hemerocallidea" in lines 301 and Elephantorrhiza elephantina in 304 cannot be abbreviated because it starts the sentence.

Reviewer #1

Comment: Both aims of the study should be improved as suggested by Reviewer #1. 

Response: The aim of the study has been revised to concisely describe the focus of the study [see page 3; line 77-78].

Reviewer #1:No

Response: Thank you for the comment. This study followed well-accepted ethnobotanical guidelines for data collection, result presentation. 

Leonti, M., 2022 The relevance of quantitative ethnobotanical indices for ethnopharmacology and ethnobotany Journal of Ethnopharmacology, 115008

Heinrich, M., Lardos, A., Leontic, M., Weckerle, C., and Willcox, M., with the ConSEFS advisory group, 2018. Best practice in research: Consensus Statement on Ethnopharmacological Field Studies, ConSEFS Journal of Ethnopharmacology 211.

Comment: This is a very minor regional ethnobotanical survey presenting a mere list of plant uses with some nonsense quantitative attempts.

The research does not the start from robust objectives (the first one is solely descriptive, while the second too large and not addressed actually by the paper), the methodology is not well described (esp. sampling) and no serious data analysis and data interpretation has been conducted. 

Response: We humbly disagree that this study is a minor regional ethnobotanical survey. The study was conduct to address an important knowledge gap which currently have global impact. Some of the medicinal plants we are reporting are also found in other regions and is a further evidence in the similarity across different ethnic groups. It is important to highlight that the documentation of medicinal plants is widely recognized as one of seven (7) priorities for strategic action in plant sciences (Cane et al 2019). From medothogical perspective, this study followed standard ethnobotanical guidelines on how to conduct and report the ethnobotanical surveys (Leonti 2022; Heinrich et al 2018). 

• Crane PR, Ge S, Hong D-Y, Huang H-W, Jiao G-L, Knapp S, et al. The Shenzhen declaration on plant sciences-Uniting plant sciences and society to build a green, sustainable Earth. Plants, People, Planet. 2019;1(1):59-61.

• Leonti, M., 2022 The relevance of quantitative ethnobotanical indices for ethnopharmacology and ethnobotany Journal of Ethnopharmacology, 115008.

• Heinrich, M., Lardos, A., Leontic, M., Weckerle, C., and Willcox, M., with the ConSEFS advisory group, 2018. Best practice in research: Consensus Statement on Ethnopharmacological Field Studies, ConSEFS Journal of Ethnopharmacology 211.

In this study, we have detailed how both of the 101 participants were selected based on the selection of the two groups of participants (traditional health practitioners and herbal vendors), which is included in the sample technique. Details of the inclusion and exclusion criteria are described in the revised manuscript. 

Reviewer #3:The manuscript entitled, "Ethnobotanical survey of medicinal plants used by indigenous knowledge holders to manage healthcare needs in children" focuses on the exploration of the medicinal plants used for treating childhood diseases in some areas of South Africa. I think the topics is interesting and matched with the journal’s scope. The authors have also presented the results of original research. However, I believe that some suggestions below might improve the manuscript before acceptance:

Response: Thank you the comments, all of the suggestions from the reviewers were taken into account and and implemented throughout the manuscript

Comment: Abstract is well written; All key findings have been presented. However, the author is suggested to mention the most cited plant family used to treat childhood diseases (Asteraceae) as this family also appears in the keywords.

Response: The plant family that appears both on the results and keywords “Asteraceae” have been removed from the keywords [see page 2;line 42]

Comment: The authors have shown very comprehensive information in the introduction section. The objective of the study should be integrated in the paragraph (not per point as presented in the MS draft).

Response: The objective of the study have been integrated in the paragraph (not per point as presented in the MS draft [ see page 3; line 77-78]

Comment: The author used standard ethnobotanical study methods. Please give number of each formula used in the study.

Response: Thanks for the observation, the standard ethnobotanical study methods, we have given the number to the all the formulas [see page 6-7; line 160-174].

Comment: The author should re-arrange the title of the table. Some information (i.e. abbreviations etc.) should be placed below the table.

Response: Thanks, for the comments, we have re-arranged the title of the tables and the abbreviations are placed below the tables [see page 12 and 17].

Comment: Author has compared the results with other ethnobotanical studies. However, most of the references described the study from Africa. It might also better to compare the results with another region outside Africa. I believe it might enrich information obtained in the study.

Response: There are several studies besides the ones from Africa, which we have compared our study with. For example, we included the following articles which focused on countries (e.g. Pakistan, Brazil and Iran) outside Africa.

• Alamgeer, Younis W, Asif H, Sharif A, Riaz H, Bukhari IA, et al. Traditional medicinal plants used for respiratory disorders in Pakistan: a review of the ethno‑medicinal and pharmacological evidence. Chinese Medicine 2018;13:48.

• Beltreschi L, de Lima RB, da Cruz DD. Traditional botanical knowledge of medicinal plants in a “quilombola” community in the Atlantic Forest of northeastern Brazil. Environment, Development and Sustainability 2018;21:1185–203.

• Beyra Á, León M, Iglesias E, Ferrándiz D, Herrera R, Volpato G, et al. Estudios etnobotánicos sobre plantas medicinales en la provincia de Camagüey (Cuba). Anales del jardín botánico de Madrid. 2004;61:185-204.” 

• Noorhosseini SA, Fallahi E, Damalas CA. Promoting cultivation of medicinal and aromatic plants for natural resource management and livelihood enhancement in Iran. Environment, Development and Sustainability 2019;http://doi.org/10.1007/s10668-019-00368-7.

Comment: It might also be good to state the importance of plant medicinal conservation in the conclusion section. Even though, the results showed that medicinal plants reported in this study are not under severe conservation strains. 

Response: Thanks for the suggestion. However, the importance of medicinal conservation has already been articulated in the study (please see page 26). This study further indicates that several studies have indicated the significance of the conserving the flora and fauna (Moyo and Van Staden, 2014; Mugomeri et al., 2016).

Comment: The authors used up to date references.

Response: Thank you for all comments and suggestions which has heped in improving the manuscript.

NB: We have also made extensive corrections, which are highlighted in red throughout the manuscript.

References 

Constitution of the Republic of South Africa., 2007. Republic of South Africa Traditional Health Practitioners Act of 2007. Cape Town, South Africa 

Moyo, M., Van Staden, J., 2014. Medicinal properties and conservationof Pelargonium sidoides DC. Journal of Ethnopharmacology 152, 243–255.

Mugomeri, E., Chatanga, P., Raditladi, T., Makara, M., Tarirai, C., 2016. Ethnobotanical study and conservation status of local medicinal plants: Towards a repository and monograph of herbal medicines in Lesotho. African Journal of Traditional, Complementary and Alternative Medicines 13, 143-156.

Stats SA, 2018. Provincial profile: North West Community Survey 2016, in: Maluleke, R. (Ed.) Provincial profile: North West / Statistics South Africa. Statistics South Africa, Pretoria, South Africa, pp. 1-93.

---

## [Editor Report · Decision Letter 1]

8 Feb 2023

Ethnobotanical survey of medicinal plants used by indigenous knowledge holders to manage healthcare needs in children

PONE-D-22-19573R1

Dear Dr. Aremu,

We’re pleased to inform you that your manuscript has been judged scientifically suitable for publication and will be formally accepted for publication once it meets all outstanding technical requirements.

Kind regards,

Branislav T. Šiler, Ph.D.

Academic Editor

PLOS ONE
---

## [Editor Report · Acceptance letter]

17 Mar 2023

PONE-D-22-19573R1 

Ethnobotanical survey of medicinal plants used by indigenous knowledge holders to manage healthcare needs in children 

Dear Dr. Aremu:

I'm pleased to inform you that your manuscript has been deemed suitable for publication in PLOS ONE. Congratulations! Your manuscript is now with our production department. 

Kind regards, 

on behalf of

Dr. Branislav T. Šiler 

Academic Editor

PLOS ONE